# Towards More Realistic Neural Network Uncertainties

## Abstract

Statistical models are inherently uncertain. Quantifying or at least upper-bounding their uncertainties is vital for safety-critical systems. While standard neural networks do not report this information, several approaches exist to integrate uncertainty estimates into them. Assessing the quality of these uncertainty estimates is not straightforward, as no direct ground truth labels are available. Instead, implicit statistical assessments are required. For regression, we propose to evaluate *uncertainty realism*—a strict quality criterion—with a Mahalanobis distance-based statistical test. An empirical evaluation reveals the need for uncertainty measures that are appropriate to upper-bound heavy-tailed empirical errors. Alongside, we transfer the variational U-Net classification architecture to standard supervised image-to-image tasks. It provides two uncertainty mechanisms and significantly improves uncertainty realism compared to a plain encoder-decoder model.

## 1 Introduction

Having attracted great attention in both academia and digital economy, deep neural networks (DNNs) (Goodfellow et al., 2016) are about to become vital components of safety-critical systems. Applications like early diagnoses of severe diseases and safe automated transport seem to be within reach (Liu et al., 2014; Bojarski et al., 2016). To actually deliver on these promises, the considerable potential of such safety-critical systems to harm humans and to cause severe damages has to be minimized. This fact comes with new challenges for the development of DNNs: next to the performance itself, further requirements such as low latency and high robustness gain in importance. Furthermore, safety-critical systems do not tolerate failures and hence have to be monitored and assessed at runtime to ensure safe functioning.

One such mean of understanding the state of a software system is measuring the statistical uncertainty of the system module given the current input. Quantifying such uncertainties helps to make decisions especially in situations of partial availability of the relevant information. In particular in modular systems, subsequent modules should profit from the addition of knowledge about the uncertainty within the processing of the current input.

Amongst others, Monte Carlo (MC) dropout and variational inference are promising approaches to estimate the prediction uncertainty of DNNs (see section 2 for more details). These approaches try to estimate more realistically the statistical uncertainties of DNNs that go beyond computing dispersion metrics on the DNN's softmax output which is known to be rather easily fooled by adversarial perturbations (Nguyen et al., 2015).

For uncertainty estimates to be used in safety-critical systems, we require them to be realistic (Horwood et al., 2014), i.e. we require these estimates to resemble the residuals (the fitting errors) of the neural network outcomes (Figure 1). This poses a conceptual challenge as standard optimization schemes do not allow for direct training of realistic uncertainties. Therefore, high predictive performance and uncertainty realism might be largely unrelated to one another. It is desirable to achieve these two objectives at the same time.

For regression, we put forward an evaluation scheme to test for uncertainty realism. For classification, we argue that existing assessment methods already (partly) satisfy these

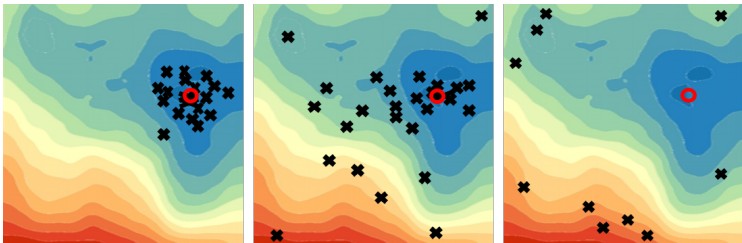

Figure 1: Symbolic image illustrating uncertainty realism. The estimated uncertainties (contour lines) are too broad (left), realistic (middle), too narrow (right) compared to the deviations of model outputs (black crosses) from ground truth (red circle).

realism demands. Instead we propose a novel approach to variational inference that provides more realistic uncertainty estimates compared to existing approaches.

In detail, our contribution is as follows:

- We propose a statistical test that allows to evaluate the realism for uncertainty mechanisms in regression tasks. These test outcomes are empirically analyzed for a 4D regression task in the vision domain (object detection with SqueezeDet and MC dropout). Further analyses call for uncertainty measures that are appropriate to upper-bound heavy-tailed empirical errors.
- We introduce a probabilistic U-Net-like FRRN semantic segmentation network and systematically assess the realism of the two uncertainty mechanisms it naturally provides. We find probabilistic FRRN to significantly improve uncertainty realism compared to (plain) FRRN.

## 2 Related work

In literature, it is common to distinguish uncertainties by their source: data-inherent uncertainty is called *aleatoric uncertainty* (e.g. rolling a dice) and uncertainty due to an limited amount of training data is called *epistemic uncertainty* (Der Kiureghian & Ditlevsen, 2009; Senge et al., 2014; Kendall & Gal, 2017; Depeweg et al., 2017). This paper focuses on epistemic uncertainty.

An approach that takes epistemic uncertainty into account are Bayesian neural networks (BNNs) (MacKay, 1992) which represent the weights by random variables. Learning a BNN requires to calculate conditional distributions of the weights given the data. Already in early applications, approximate solutions are used (MacKay, 1992; Neal, 1993). Many recent applications of BNNs to deep architectures are based on variational approximations to Bayesian inference (Attias, 2000), e.g. MC dropout (Gal & Ghahramani, 2016), early stopping (Duvenaud et al., 2016) and weight decay (Blundell et al., 2015). For our experiments, we apply MC dropout to a regression problem in the detection setting. MC dropout was already used for different regression tasks (Kendall & Gal, 2017) and also in the detection setting (Miller et al., 2018).

Bayesian frameworks are also used in other settings than BNNs: the *Probabilistic U-Net* (Kohl et al., 2018), a model for segmentation, uses a neural Network as a feature extractor, the weights of which are not treated as random variables. To obtain a distribution over possible segmentations, it is assumed that there are hidden random variables that generate a distribution over the output space. The probabilistic U-Net was introduced to model an aleatoric uncertainty, namely the label ambiguity caused by different ground truth segmentations. The approximations mentioned above both represent the output by a sample from the approximate posterior distribution. The approximations make it necessary to assess the quality of the uncertainty measures.

One way to assess the quality of an uncertainty estimate is critically studying the algorithm itself. For example Osband (2016) points out that MC dropout (Gal & Ghahramani, 2016)—

unlike a corresponding BNN—does not converge to concentrated distributions in the infinite data limit (see also Gal et al. (2017)) and Bishop (2006) shows that variational approaches tend to underestimate the variance. An alternative way is to treat the uncertainty mechanism as a black box and assess how well the estimated uncertainties fit to the reality. Lacking a ground truth label for the uncertainty, this comparison has to be indirect (Lakshminarayanan et al., 2017). The assessment approaches differ between classification and regression tasks.

**Classification** For the classification task, it is typical to determine an uncertainty score (either directly on the network output (Gal & Ghahramani, 2016; Hendrycks & Gimpel, 2017) or by an additional model (Oliveira et al., 2016)) and to raise a flag if this score surpasses a threshold. Given an approximation to a posterior distribution (or a sample from it) it is possible to derive **scores** based on the expected softmax and higher moments. Typical such scores on the expected softmax are the entropy (Mukhoti & Gal, 2018) or the maximum (Hendrycks & Gimpel, 2017). Scores based on the second moment are the variance (of the winning class in sampling or of the expected softmax) and mutual information (MI) (Mukhoti & Gal, 2018).

To make evaluation of the above scores independent of a threshold value, areas under the receiver operating characteristic (AUROC), the precision-recall curve (AUPRC) or the negative predictive value versus the recall curve are considered (Hendrycks & Gimpel, 2017; Mukhoti & Gal, 2018). Alternatively, one can assess the calibration quality of the resulting softmax (Lakshminarayanan et al., 2017; Guo et al., 2017; Gal et al., 2017).

**Regression** For regression tasks, it is suitable to represent the uncertainty of an approximation to the posterior distribution (or of an sample from it) by its (empirical) covariance matrix. A very strict way to assess this matrix is the application of a suitable statistical test, as it is in astronautics (*covariance realism*, see Horwood et al. (2014) for details). A less strict metric is a multidimensional extension of the standardized mean-squared error (SMSE) (Rasmussen, 2003) that was used by Fruehwirt et al. (2018) to assess a one-dimensional regression model based on MC dropout (this work also covers other methods than MC dropout). More frequently, models are assessed by the average log-likelihood (Blei et al., 2006; Walker et al., 2016; Gal & Ghahramani, 2016).

The methods described above assess the complete covariance matrix. For further assessments this matrix is reduced to a **score** measuring a certain aspect of the corresponding ellipsoid. Plausible scalar **scores** include the determinant, the maximal eigenvalue or the maximal diagonal entry of the matrix. A completely different method of uncertainty estimate assessment is investigating its behavior. For example Kendall & Gal (2017) check if the epistemic uncertainty decreases when increasing the training set or Wirges et al. (2019) study how the level of uncertainty depends on the distance of the object for some 3D environment regression task.

## 3 Towards more realistic neural network uncertainties

An uncertainty mechanism should be realistic. To assess its realism we need suitable metrics. The metrics should help us to decide if—or to which degree—the uncertainty reflects the actual statistical weaknesses of the model. Earlier attempts to uncertainty realism can be found in Mukhoti & Gal (2018). For regression tasks, we derive a mathematical criterion based on Mahalanobis distances. For classification, existing assessment methods already allow to judge how well uncertainties point out the weaknesses of the model. Here, we propose a novel approach to variational inference that provides more realistic uncertainty estimates compared to existing approaches.

**Regression**

For regression tasks, uncertainty mechanisms like MC dropout produce an output sample instead of a single output. The sample covariance matrix is a measure for the network uncertainty. In other words we fit a Gaussian to the sample of predictions and call this Gaussian the posterior predictive distribution. There are multiple ways to assess how good this posterior predictive distribution fits to test data. The most common way is to calculate the negative log-likelihood of the test data. However, test log-likelihood is a combined

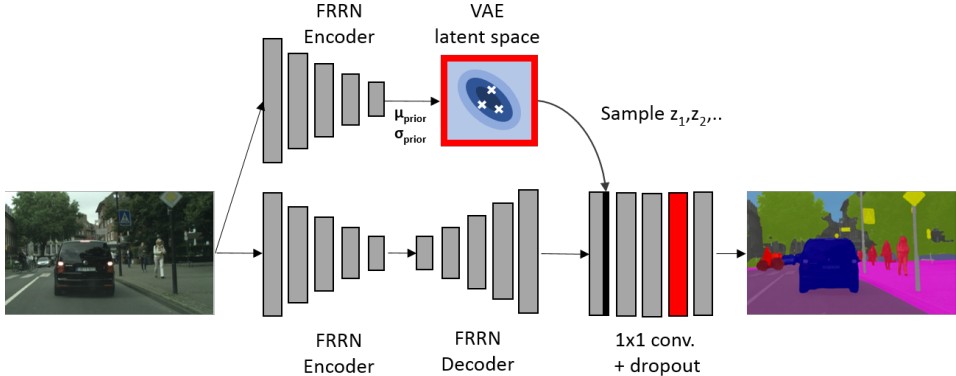

Figure 2: Architecture of the proposed variational FRRN (at inference). The two uncertainty mechanisms *latent space sampling* and *MC dropout* are highlighted in red.

evaluation metric for means and variances and thus it is not ideally suited for analyzing the quality of the estimated variances. Instead, it is preferable to assess the variances separately. We propose to do this by means of a statistical test:

For conceptual clarity, we consider a simple 1D regression task for the moment. A neural network with MC dropout is trained to predict $y_{i,gt}$ from $x_i$. At test time, the network generates—for each input $x_i$—a sample $\{y_{i,k}\}$ that is summarized as $(\mu_i, \sigma_i)$. The network residual is $\xi_i = y_{i,gt} - \mu_i$. Here, uncertainty realism boils down to requiring: $\xi_i \sim \mathcal{N}(0, \sigma_i)$. As only one $\xi_i$ is available for a given $x_i$, rewriting this criterion as

$$\frac{\xi_i^2}{\sigma_i^2} = \frac{(y_{i,gt} - \mu_i)^2}{\sigma_i^2} \sim \chi^2(d = 1) \tag{1}$$

allows to actually test it—independent of a specific dataset. Note that not only unrealistic uncertainties but also regression bias could lead to violations of this condition.

Generalizing this statistical criterion from 1D to a higher dimensional regression task, the covariance realism can be assessed using the squared Mahalanobis distance (Horwood et al., 2014):

$$M^2_{\boldsymbol{\mu}_i, \boldsymbol{\Sigma}_i}(\mathbf{y}) = (\mathbf{y} - \boldsymbol{\mu}_i)^T \, \boldsymbol{\Sigma}_i^{-1} (\mathbf{y} - \boldsymbol{\mu}_i) \, ,$$

where $\boldsymbol{\mu}_i$ and $\boldsymbol{\Sigma}_i$ are derived from the uncertainty mechanism for data point $\mathbf{x}_i$. If $\boldsymbol{y}$ is sampled from a Gaussian distribution with mean $\boldsymbol{\mu}$ and covariance matrix $\boldsymbol{\Sigma}$, $M^2_{\boldsymbol{\mu}, \boldsymbol{\Sigma}}(\boldsymbol{y})$ follows a chi-squared distribution $\chi^2(d)$, where $d$ is the dimension of $\boldsymbol{y}$. Note that this statement holds for every choice of $\boldsymbol{\mu}$ and $\boldsymbol{\Sigma}$ and hence also in our setting, where each observation $\boldsymbol{y}_{i,gt}$ is assumed to be a realization of a Gaussian with different parameters $\{\boldsymbol{\mu}_i, \boldsymbol{\Sigma}_i\}$. Hence, if the estimates $\boldsymbol{\mu}_i$ and $\boldsymbol{\Sigma}_i$ are realistic, a strict uncertainty realism criterion is given by requiring the set $\mathcal{M}_{\text{gt}} = \{M^2_{\boldsymbol{\mu}_i, \boldsymbol{\Sigma}_i}(\boldsymbol{y}_{i,gt})\}$ to follow a $\chi^2(d)$-distribution.

**Classification**

For classification tasks, not only the study of softmax variances but also the study of the mean softmax is interesting because it provides (inter-class) uncertainty information. This is in contrast to regression tasks. To treat these different uncertainty sources in a uniform way, we derive scalar scores from them, e.g. entropy or variance.

The practical use of such an uncertainty score is to identify inputs for which the model is wrong. The better we are in making this decision, the more realistic is the uncertainty score. This realism is quantified by AUC values.

A standard uncertainty mechanism for classification tasks is MC dropout (Gal & Ghahramani, 2016) as an approximation of a BNN. However, it makes use of an uninformed prior over the weights. We propose an alternative mechanism by studying variational inference-based uncertainty inspired by the probabilistic U-Net (Kohl et al., 2018). In contrast, our

approach of using the variational architecture (Figure 2) is to focus less on creating different interpretations of an input image, but to treat the latent space as a more sophisticated and focused mechanism to encode data-inherent uncertainty. Furthermore, instead of using a U-Net as the base structure for the segmentation task and the encoders, we use an FRRN-based (Pohlen et al., 2017) architecture (see appendix A for details). Additionally, we allow to combine the latent space sampling with MC dropout by inserting a dropout layer right before the last convolution of the network. To the best of our knowledge, this architecture yields the first DNN that allows for evaluation and fusion of the two mentioned uncertainty mechanisms.

## 4 EMPIRICAL EVALUATION

Following the outlined assessment scheme for neural network uncertainties, we evaluate their realism for the regression task of object detection. Moreover, we analyze the realism of the two uncertainty mechanisms of the novel variational FRRN for semantic segmentation.

### 4.1 REGRESSION - OBJECT DETECTION

We consider SqueezeDet (Wu et al., 2017), a lightweight single-stage detector network, that uses a pre-trained SqueezeNet as its backbone. It is trained on KITTI (Geiger et al., 2012) and returns a four-tuple ($d = 4$) of center coordinates, width and height of a 2D bounding box for detected objects. The uncertainty mechanism we use is MC dropout ($p = 0.5$) before the last convolutional layer. The MC sampling of the dropout layer is done before the thresholding and non-maximum suppression stage. The output sample $\{\boldsymbol{y}_{i,k}\}$ for a given input $\boldsymbol{x}_i$ with ground truth $\boldsymbol{y}_{i,gt}$ is characterized by its mean $\boldsymbol{\mu_y}$ and its covariance $\boldsymbol{\Sigma_y}$. Following section 3, we calculate the squared Mahalanobis distance $M^2_{\boldsymbol{\mu}_i, \boldsymbol{\Sigma}_i}(\boldsymbol{y}_{i,gt})$. $\mathcal{M}_{\text{gt}} = \{M^2_{\boldsymbol{\mu}_i, \boldsymbol{\Sigma}_i}(\boldsymbol{y}_{i,gt})\}$ denotes the set of squared Mahalanobis distances for the entire test set $\{\boldsymbol{x}_i, \boldsymbol{y}_{i,gt}\}$. To apply the strict uncertainty realism criterion, we test if this set is drawn from $\chi^2(d = 4)$. This yields a p-value close to zero and therefore we have to reject the hypothesis and find the uncertainty mechanism to be unrealistic: MC dropout does not provide realistic uncertainty estimates in this empirical setting. In contrast, the set of intra-sample squared Mahalanobis distances $\mathcal{M}_{\text{sample}} = \{M^2_{\boldsymbol{\mu}_i, \boldsymbol{\Sigma}_i}(\boldsymbol{y}_{i,k})\}$ is—in accordance with theory—$\chi^2$-distributed. A comparison of these distributions is shown in Figure 3 (left panel, for per-component visualizations see Figure 9 in the appendix). The width of the MC sample distribution is one order of magnitude smaller than the actual estimation errors - highlighting that variational approaches underestimate variances (Bishop, 2006). Handling this deviation in an ad-hoc fashion by scaling the variance of $\mathcal{M}_{\text{sample}}$ to match the variance of $\mathcal{M}_{\text{gt}}$ is shown in Figure 3 (middle panel). The higher moments of $\mathcal{M}_{\text{gt}}$ are still deviating and cause a decay that is slower than exponential, i.e. a fat tail.

While Mahalanobis distances allow for a combined assessment of the realism of the size and orientations of the covariance ellipsoid, an isolated assessment of the covariance orientation can be done by analyzing the angle $\alpha = \angle(\boldsymbol{v}^{\max}_{\boldsymbol{\Sigma}}, \boldsymbol{\mu_y} \text{-} \boldsymbol{y}_{gt})$ enclosed by the largest eigenvector of the covariance $\boldsymbol{v}^{\max}_{\boldsymbol{\Sigma}}$ and the estimation error $\boldsymbol{\mu_y} \text{-} \boldsymbol{y}_{gt}$ (Figure 3, right panel). The high resemblance of the angle distribution with the distribution of the differential solid angle of the 3-sphere emphasizes that no covariance orientation realism is given.

Following section 3, we check if MC dropout is—at least—a good indicator for realism, i.e. if the mean estimation error increases monotonically with the uncertainty score $\boldsymbol{\Sigma_y}$. Here, we approximate the covariance with its determinate $\det(\boldsymbol{\Sigma_y})$ and its largest component $\Sigma^{\max}$ as rough scalar measures of its size. Figure 4 (left and right panel) suggests that such a monotonicity is given. Due to the discussed fat tails, however, upper-bounding estimation errors by uncertainty estimates cannot be done using multiples of the standard deviation. The empirical 99% quantiles (Figure 4, green) are underestimated by the—in case of normality—corresponding $2.576\sigma$-interval (red), partly by far. Therefore, residual risks should be quantified using appropriate tail measures such as quantiles or tail mean values.

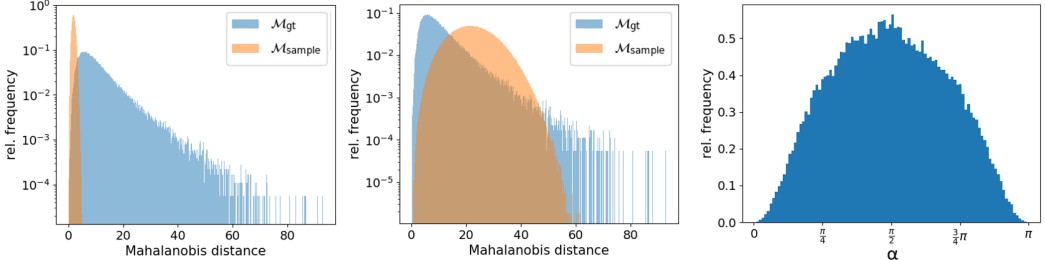

Figure 3: Assessment of uncertainty realism for the regression task. Left: empirical distributions $\mathcal{M}_{\text{sample}}$ and $\mathcal{M}_{\text{gt}}$, middle: rescaled $\mathcal{M}_{\text{sample}}$ to match the variance of $\mathcal{M}_{\text{gt}}$, right: distribution of angles $\alpha$ enclosed by the error direction and the covariance ellipsoid orientation.

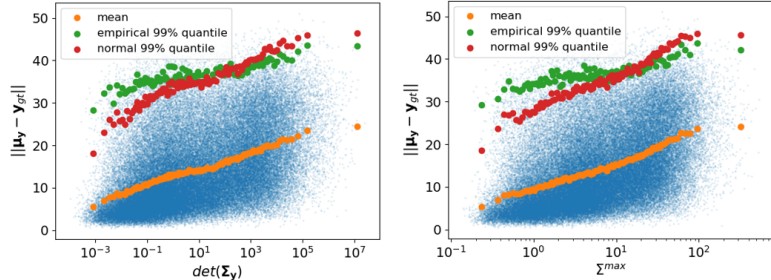

Figure 4: Uncertainty scores and estimation errors (left: covariance determinant, right: largest covariance element).

## 4.2 Classification - semantic segmentation

The proposed variational FRRN (Figure 2, appendix A.2) is trained for $3 \times 10^5$ steps on the Cityscapes dataset (Cordts et al., 2016) with a batch size of 4 and a drop rate of $p = 0.5$. The experiments are conducted on the images of the city "Münster". In order to compare the three uncertainty mechanisms that the variational FRRN provides, each test set image is processed three times: with only MC dropout switched-on (MC), with only latent space sampling switched-on (CVAE) and with both mechanisms switched-on (CVAE + MC). Regardless of the chosen uncertainty mechanism, the number of samples is fixed to 50. As uncertainty scores we consider the highest probability and the entropy of the mean softmax, the variance of the winning class and the MI (see section 2). To evaluate the realism of the mechanisms not only within the training data distribution but also out-of-distribution, images are vertically flipped and again processed three times as described above. The uncertainty realism is measured by calculating the respective areas under the ROC (AUROC) and the precision-recall curve (AUPRC) where we take the correctly classified pixels as positive. As a baseline we use the plain softmax output without any uncertainty mechanism being switched-on.

Table 1 shows that for in-data samples the plain softmax probability is a reasonably accurate uncertainty estimation and cannot be outperformed by the advanced mechanisms. This may be related to the high accuracy of the network because good predictions require comparably less challenging uncertainty estimations. An example for an in-data prediction and the corresponding estimated uncertainty can be seen in Figure 7 in the appendix.

For out-of-data images, however, the sampling mechanisms allow a significant improvement. Sampling with the variational FRRN yields the most realistic uncertainties with an AUROC value of 0.72 when using the entropy score and an AUPRC of 0.755 when using the mean softmax probability. The influence of the different scores on the assessment is generally small. Fusing latent space sampling with MC dropout does not lead to a further improvement.

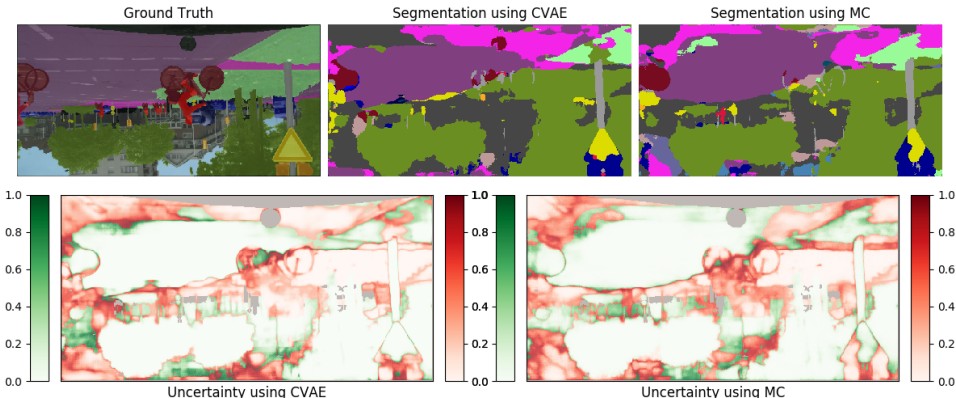

Figure 5: Segmentations and uncertainty maps produced by variational FRRN. Segmentations: ground truth (top left), FRRN with CVAE sampling (top middle), FRRN with MC dropout (top right). Uncertainty maps: FFRN + CVAE sampling (bottom left), FRRN + MC dropout (bottom right). Uncertainties of correctly classified pixels are shown in green, of misclassified pixels in red.

Table 1: Realism assessment for pixel-wise classification (in-sample and out-of-sample). Compared mechanisms are the softmax baseline (none), MC sampling (MC), latent space sampling (CVAE) and a combination of the latter (CVAE + MC).

| mechanism | score | assessment | | | |
|---|---|---|---|---|---|
| | | in-sample | | out-of-sample | |
| | | AUROC | AUPRC | AUROC | AUPRC |
| none | max. comp. softmax | **0.945** | **0.997** | 0.642 | 0.643 |
| none | entropy | 0.944 | 0.997 | 0.639 | 0.640 |
| | *mean-based scores $f(\boldsymbol{\mu_y})$* | | | | |
| MC | max. comp. softmax | 0.945 | 0.997 | 0.651 | 0.653 |
| MC | entropy | 0.943 | 0.997 | 0.657 | 0.655 |
| MC | MI | 0.943 | 0.997 | 0.654 | 0.655 |
| CVAE | max. comp. softmax | 0.944 | 0.997 | 0.718 | **0.755** |
| CVAE | entropy | 0.944 | 0.977 | **0.720** | 0.751 |
| CVAE | MI | 0.926 | 0.995 | 0.717 | 0.747 |
| CVAE + MC | max. comp. softmax | 0.946 | 0.997 | 0.713 | 0.742 |
| CVAE + MC | entropy | 0.944 | 0.997 | 0.714 | 0.742 |
| CVAE + MC | MI | 0.919 | 0.996 | 0.706 | 0.735 |
| | *covariance-based scores $f(\boldsymbol{\Sigma_y})$* | | | | |
| MC | variance of max. comp. softmax | 0.934 | 0.996 | 0.657 | 0.655 |
| CVAE | variance of max. comp. softmax | 0.936 | 0.997 | 0.718 | 0.753 |
| CVAE + MC | variance of max. comp. softmax | 0.914 | 0.995 | 0.702 | 0.733 |

Figure 5 depicts the predictions and uncertainty estimations of the network given an out-of-data input image. As expected, the network fails to correctly classify the majority of the pixels. However, both applied mechanisms allow to detect a reasonable amount of the occurring misclassifications and assign a low uncertainty to regions with many true positives.

Figure 6 shows the ROC and precision-recall curves for the different mechanisms and the entropy score. It can be seen that the general shape is similar for all mechanisms. Further results for using a ROC curve-based thresholding to reject predictions with high chance of being wrong can be found in the appendix (Figure 8). Comparing the mechanisms qualitatively (Figure 5) and empirically (Figure 6, Table 1), we see that MC dropout and the latent space sampling seem to express the same kind of uncertainty. Both mechanisms generate better probability estimations for out-of-data samples compared to plain softmax.

At a first glance, the similarity between the two mechanisms is surprising because they function in different ways. However, both are based on the principles of variational inference, thus approximate the posterior by sampling from the model parameters. Fusing both

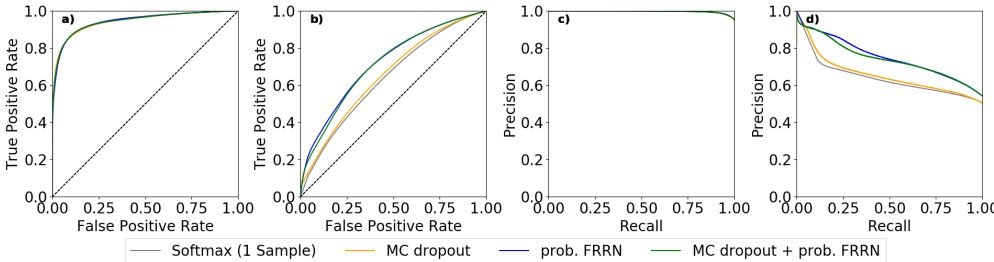

Figure 6: In-sample and out-of-sample comparison of different uncertainty mechanisms. a) and b) show in-sample and out-of-sample ROC curves, c) and d) the equivalent precision-recall curves. Class entropy is used as uncertainty score.

mechanisms leads to a different sample of parameters, but from the same distribution, which may be why we see no further improvements in the uncertainty estimate.

## 5    DISCUSSION AND FUTURE WORK

We argue to evaluate uncertainty realism in regression tasks with a Mahalanobis distance-based statistical test. We improve uncertainty realism in a classification task by using the proposed variational FRRN instead of a (plain) FRRN.

**Regression** For the considered regression task in the vision domain, MC dropout does not provide realistic uncertainty estimates as it does not fit the orientation, width or non-normality of the estimation error distribution. However, it is important to highlight that our experimental setting adds further approximations to the approximations underlying MC dropout (small network, dropout applied only to one layer). This setup is commonly used though and we emphasize to carefully validate MC dropout results for each safety-critical application. While not being realistic, we find it to be a good indicator for realism. The observed non-normality of the error distribution indicates that naive uncertainty upper-bounds on regression errors might fall short. Instead, we recommend the usage of appropriate uncertainty measures such as quantiles or tail mean values.

**Classification** When comparing the MC dropout with the latent space sampling technique, it seems that both approximate the same form of uncertainty. While small scale differences appear in examplary images, the overall estimation focuses on the same aspects and empirical evaluations show similar tendencies. Both mechanisms allow to reason about epistemic uncertainties. We state that the similarities originate from the mutual base idea of estimating the models uncertainty by approximating the posterior distribution. Compared to MC dropout, the latent space sampling seems to be the more informed uncertainty mechanism.

**Future work** Uncertainty realism might be improved by using multiple dropouts in the MC setting, adjusting loss functions and other training hyper parameters or using thresholds per class for the score. Studying other out-of-data directions (e.g. lighting conditions, fog and unknown classes) and tasks from other domains could strengthen our results. A shortcoming of probabilistic U-Net is that the latent space sample is only used as an additional channel which the network can choose to ignore. PHiSeg (Baumgartner et al., 2019) is an extension of the probabilistic U-Net that addresses this. Transferring the PHiSeg approach to our setting and studying the resulting uncertainties seems promising. Important further work is to prove residual risk bounds and to formalize requirements for uncertainty estimates of neural networks. Relating uncertainty estimates to adversarial robustness needs further research.

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

# A    ADDITIONAL IMPLEMENTATION DETAILS

## A.1    BASE FRRN

In this section we describe the FRRN base architecture that we use for the experiments in section 4.2. Initially, the input to the FRRN gets fed into a convolutional layer with a 5 x 5-kernel, followed by three residual units consisting of two convolutions with a skip connection. The data flow then splits into a pooling stream and a residual stream. While the data in the pooling stream gets down- and up-sampled to extract features, the residual stream remains on full resolution, retaining the spatial information. The two streams get combined at each scale in so called full-resolution residual units (FRRUs) by pooling the residual stream to the other streams dimensions and merging it through concatenation and two convolutions. The last upsampling and stream concatenation is followed by another three residual units and a 1 x 1 convolution to classify the pixels. Finally a softmax layer is used to calculate the class probabilities.
We start with a base number of 24 channels and double, respectively halve the number with each down- and up-sampling step. For dimension reduction or enlargement we use max pool and bilinear upscale operations. The residual stream is kept at consistently 16 channels, thus after a final concatenation we end with 40 feature maps that are used for the classification. If not stated otherwise all convolutions were run with 3 x 3-kernels and are followed by a ReLU activation.

## A.2    VARIATIONAL FRRN

Compared to the architecture of the probabilistic U-Net, we made various changes: As the base structure we use a full-resolution residual network (FRRN, see appendix A.1) and its feature extraction part and thus the downsampling operations alongside the full-resolution residual units for the variational encoders (Figure 2). To use the features of both the residual and the pooling stream for the probability estimation in the encoders, the final activations of the residual stream are pooled and concatenated with the pooling stream. The resulting final feature maps are global average pooled and a $1 \times 1$ convolution is used to predict a mean and variance. As proposed by Kohl et al. (2018) we use a dimension of six for the latent space and broadcast the drawn sample to the last activation map of the FRRN, followed by three $1 \times 1$ convolutions.

# B    ADDITIONAL RESULTS FOR THE VARIATIONAL FRRN

## B.1    QUALITATIVE RESULTS

Figure 7 shows the predictions and corresponding uncertainty estimates of the variational FRRN. Especially class boundaries and distant objects are assigned a high uncertainty. Note that the selected input image is challenging: The fence on the left side of the image is see-through. While it is labelled as "fence", the network sees the objects behind it. The resulting misclassifications are assigned a high uncertainty.

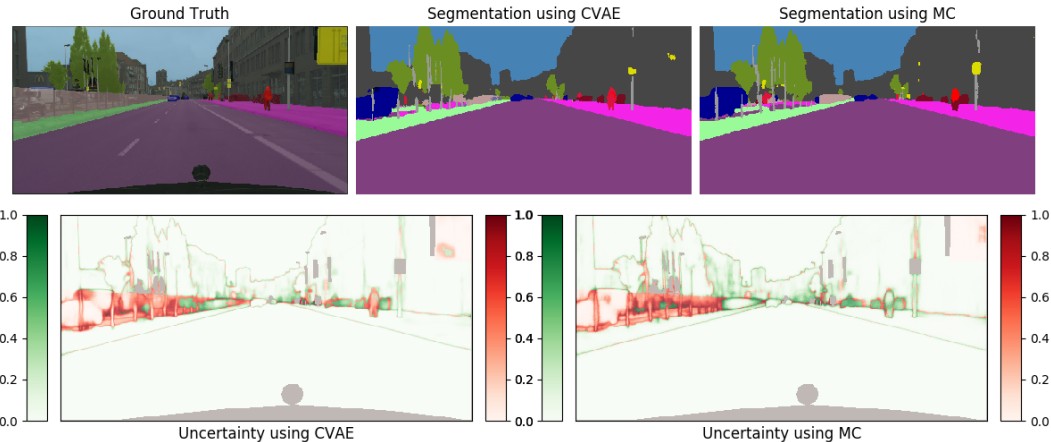

Figure 7: Uncertainty of correct and false classified pixels. Upper row: The flipped input image, overlayed with the Ground truth segmentation as well as the predictions based on the variational FRRN and MC dropout (50 samples each). The lower row shows the normalized, pixelwise uncertainty of correctly classified pixels (green) and misclassified pixels (red). Gray regions are not included in training and evaluation.

## B.2    QUANTITATIVE RESULTS

The ROC curves allow to find a threshold that maximises the trade-off between the true positive and false negative rate. Finding this threshold and applying it to the uncertainty estimations allows to reject pixels that have a high chance of being misclassified. Figure 9 shows the amount of false negatives (FN) and true positives (TP) that were detected as being "misclassified".

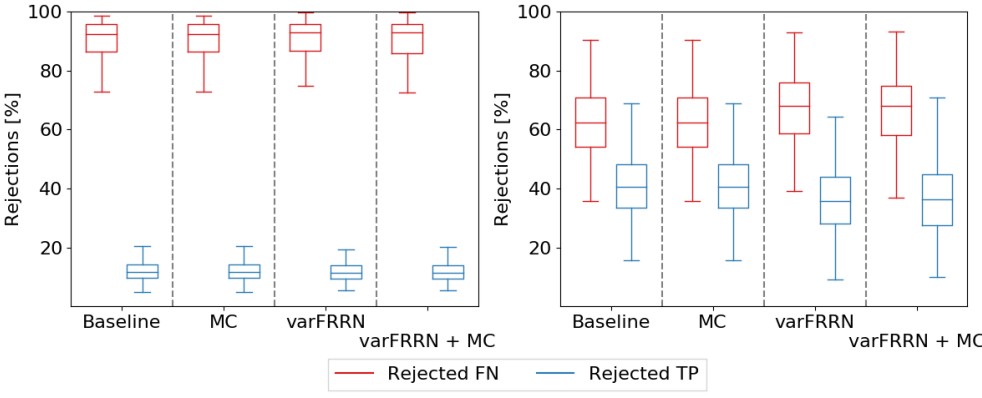

Figure 8: Rejection rates of FN and TP predictions with high uncertainty. The threshold is based on the ROC curve and the entropy score. Left: Rejections on in-data samples, right: on out-of-data samples.

## C  Extended evaluation of the uncertainty realism of SqueezeDet using MC dropout

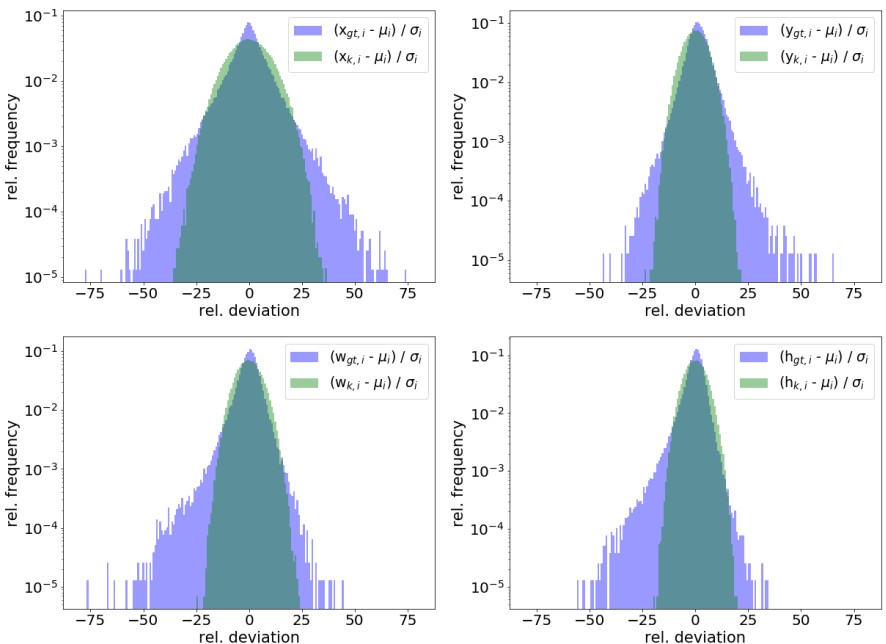

Figure 9: Assessment of uncertainty realism for the regression task. Distributions of scaled MC-dropout samples and estimation errors in x-direction (top left), y-direction (top right), for width (bottom left) and height (bottom right).

