# OpenReview forum: "Towards More Realistic Neural Network Uncertainties"
_ICLR.cc/2020/Conference — Reject_

### Official Review · AnonReviewer2 · 2019-10-16
**Official Blind Review #2**

**Rating:** 1

**Review:**

The paper a new method for evaluating prediction uncertainties for regression and classification tasks.

I argue this work should be rejected. For regression the method is based on Gaussian assumption, which has no reason to be correct (especially for BayesianNN), and for classification the method is not clear at all.

Detailed remarks:
- For a BNN, there is no reason why p(y|x) is Gaussian, so even if the residual is rejected as chi-square this doesn't mean anything.
- I am unclear as to how the classifier uncertainty is measured.
- The paper claims that "it is preferable to assess the variances separately." which I agree with, but a mistake in the mean would effect your chi-square test, so it is not separated.
- I am unclear as to what statistical test you perform to accept/reject the chi-square hypothesis.


**Experience Assessment:**

I have published one or two papers in this area.

**Review Assessment: Checking Correctness Of Derivations And Theory:**

N/A

**Review Assessment: Checking Correctness Of Experiments:**

I did not assess the experiments.

**Review Assessment: Thoroughness In Paper Reading:**

N/A

---

### Official Review · AnonReviewer1 · 2019-10-22
**Official Blind Review #1**

**Rating:** 3

**Review:**


The paper makes the following two contributions: 1) a new metric to measure the realism of uncertainty estimates for regression problems which uses a Mahalanobis distance-based statistical test. 2) a new probabilistic architecture for semantic segmentation.

Overall I do not think that the paper is qualifies for acceptance, because a) both contributions are only loosely connected and b) some parts are confusingly written or poorly motivated, making the paper hard to follow.

After reading the paper it still remains unclear to me why the proposed statistical test is superior to other popular metrics, such as the log-likelihood or the metrics proposed by Mukhoti and Gal. I think the paper would benefit from a more detailed discussion that highlights the differences between the proposed metric and other commonly used metrics.
Furthermore, in the experiments, the paper only shows that MC dropout doesn't achieve realistic uncertainty estimates. I think concluding that variational approaches underestimate the variance is a bit of stretch (see section 4.1) , i e. it would be more convincing if other approaches (e.g Blundell et al. Hernandes-Lobato and Adams) are also considered.
The paper also just states that uncertainty estimates obtained by MC dropout are unrealistic but doesn't elaborates how to improve them.

The second contribution, a probabilistic architecture for semantic segmentation, is not introduced in the main paper. Instead details are only provided in the appendix. In my opinion the paper would be easier to follow if it would contain a section that motivates and described the proposed architecture before jumping directly to the experiments.

Minor comments:

- the acronym FRRN is not defined

- Section 4.2 last paragraph: I don't understand why samples obtain by MC dropout and from the latent space are considered to be from the same distribution? While both methods approximate the weight posterior they use different approximations for that.

- In the introduction the paper states that the proposed segmentation network is a U-net like FRRN architecture , however in section 3 it says instead of a U-net based architecture a FRRN based architecture is used. This is somewhat confusing.

- Section 4.1: how are the variances scaled for Figure 3 middle?

- Section 4.1 It would be also interesting if other metrics, such as log-likelihood or RMSE, to see how well the model is able to fit the data.

- I am also surprised that CVAE + MC underperforms to just using CVAE?

- Section 4.1: how are the variances scaled for Figure 3 middle?

References:

Evaluating Bayesian Deep Learning Methods for Semantic Segmentation
Jishnu Mukhoti, Yarin Gal
arXiv:1811.12709 [cs.CV]

Weight Uncertainty in Neural Networks
Charles Blundell, Julien Cornebise, Koray Kavukcuoglu, Daan Wierstra
ICML 2015


Hernández-Lobato J. M. and Adams R.
Probabilistic Backpropagation for Scalable Learning of Bayesian Neural Networks,
ICML 2015

**Experience Assessment:**

I have read many papers in this area.

**Review Assessment: Checking Correctness Of Derivations And Theory:**

N/A

**Review Assessment: Checking Correctness Of Experiments:**

I assessed the sensibility of the experiments.

**Review Assessment: Thoroughness In Paper Reading:**

N/A

---

### Official Review · AnonReviewer3 · 2019-10-23
**Official Blind Review #3**

**Rating:** 1

**Review:**

This paper considers the important open question of how to ensure that uncertainty estimates of neural network predictions actually reflect real world error distributions. The paper introduces an uncertainty quality metric along with a statistical test based off this metric that enables a binary decision of whether to accept a model’s uncertainties as realistic. The paper also introduces a new model for supervised image-to-image tasks that combines two existing uncertainty mechanisms, and achieves reasonable uncertainty estimates, in particular, demonstrating robustness to out-of-sample data.

I am tending to reject, because although each of the two distinct contributions are good starts on interesting approaches, neither provides a convincing solutions for the main question, and the two contributions are quite distinct, so that the paper lacks a consistent thread.

First, the Mahalanobis-based uncertainty evaluation makes sense, but it is not clear what it adds beyond the standard average negative log-likelihood (NLL) metric. Mahalanobis distance increases monotonically with negative log-likelihood, so is there any reason to expect it is a better way to compare models based on their uncertainty? Is there some experiment that could show that the new metric is better than NLL at evaluating uncertainty realism?

The statistical test could potentially be a nice real world tool for deciding whether to trust neural network uncertainties. However, the paper only applies the test in one case, where the MC dropout model is shown to have unrealistic uncertainties. To show that this test is useful, there should also be experiments where uncertainty estimates are shown to be realistic. Is there some alternative to MC dropout for regression or some improvements to the algorithm that could yield realistic uncertainties under this statistical test? E.g., would the U-Net-based model in the paper pass the test if it were adapted to regression problems?

Second, the new U-Net-based classification model looks like a reasonable approach, but it is disjoint from the new statistical test, and it is not clear that the method yields improvements to uncertainty realism, since there are no comparisons to external results. Since only a new architecture is used, it is not clear that the deficiencies in uncertainty realism are not architecture-specific. Is it that there are no existing architectures that can be applied to this problem? Also, is there some more realistic setting where the CVAE approach would improve in-sample scores, i.e., where out-of-sample data is not generated synthetically? Can the statistical test be adapted to the classification setting? Similar, to the case of regression, the uncertainty realism metrics used for classification are tightly coupled with the prediction accuracy; is there a way to decouple these, would one want to? The conclusion that CVAE is better than MC for this problem is solid, but is there a more general conclusion to be drawn? E.g., could a CVAE model yield realistic estimates in the regression setting?


**Experience Assessment:**

I have published one or two papers in this area.

**Review Assessment: Checking Correctness Of Derivations And Theory:**

I assessed the sensibility of the derivations and theory.

**Review Assessment: Checking Correctness Of Experiments:**

I assessed the sensibility of the experiments.

**Review Assessment: Thoroughness In Paper Reading:**

I read the paper at least twice and used my best judgement in assessing the paper.

---

### Decision · Program_Chairs · 2019-12-19

**Decision:**

Reject

**Comment:**

This paper proposes two contributions to improve uncertainty in deep learning.  The first is a Mahalanobis distance based statistical test and the second a model architecture.  Unfortunately, the reviewers found the message of the paper somewhat confusing and particularly didn't understand the connection between these two contributions.   A major question from the reviewers is why the proposed statistical test is better than using a proper scoring rule such as negative log likelihood.  Some empirical justification of this should be presented.